# Expression Patterns of MYB (V-myb Myeloblastosis Viral Oncogene Homolog) Gene Family in Resistant and Susceptible Tung Trees Responding to *Fusarium* Wilt Disease

**Xue Wang** [1,2]**, Qiyan Zhang** [1,2]**, Ming Gao** [1,2]**, Liwen Wu** [1,2]**, Yangdong Wang** [1,2] **and Yicun Chen** [1,2,*]

[1]    State Key Laboratoty of Tree Genetics and Breeding, Chinese Academy of Forestry, Bejing 100091, China; wangxueale@163.com (X.W.); zhangqy@caf.ac.cn (Q.Z.); 4862705@163.com (M.G.); wuliwenhappy@caf.ac.cn (L.W.); wyd11111@126.com (Y.W.)

[2]    Institute of Subtropical Forestry, Chinese Academy of Forestry, Hangzhou 311400, China

[*]    Correspondence: chenyc@caf.ac.cn; Tel.: +86-571-6332-7982

**Abstract:** *Vernicia fordii* (tung oil tree) is famous in the world for its production of tung oil. Unfortunately, it was infected by the soil-borne fungus *Fusarium oxysporum f.* sp. *fordii* 1 (*Fof*-1) and suffered serious wilt disease. Conversely, its sister species *V. montana* is highly resistant to *Fof*-1. The MYB (v-myb myeloblastosis viral oncogene homolog) transcription factors were activated during the pathogen *Fof*-1 infection according to our previous comparative transcriptomic results. Depending on whether the sequence has a complete MYB-DNA-binding domain, a total of 75 *VfMYB* and 77 *VmMYB* genes were identified in susceptible *V. fordii* and resistant *V. montana*, respectively. In addition, we detected 49 pairs of one-to-one orthologous *Vf/VmMYB* genes with the reciprocal-best BLAST-hits (RBH)method. In order to investigate the expression modes and the internal network of MYB transcription factors in the two species responding to *Fusarium* wilt disease, the expressions of *Vf/VmMYBs* were then investigated and we found that most orthologous *Vf/VmMYB* genes exhibited similar expression patterns during the *Fof*-1 infection. However, four pairs of *Vf/VmMYB* genes, annotated as unknown proteins and mediator of root architecture, demonstrated absolute opposite expression patterns in the two *Vernicia* species responding to *Fof*-1. The interaction network of *VmMYB* genes were further constructed using weighted gene co-expression network analysis (WGCNA) method and four hub genes showing extremely high interaction with the other 1157 genes were identified. RT-qPCR result verified the opposite expression pattern of the hub gene *VmMYB011* and *VmMYB041* in two *Vernicia* species. In summary, co-expression network of the *Vf/VmMYBs* and significantly opposite related pairs of genes in resistant and susceptible *Vernicia* species provided knowledge for understanding the molecular basis of *Vernicia* responding to *Fusarium* wilt disease.

**Keywords:** MYB; transcription factors; *Fusarium* wilt disease; tung oil tree; expression pattern

## 1. Introduction

Tung trees (*Vernicia*) are world-famous woody non-edible oil plants that play an important role in industry, biology, energy, medicine and ecological environment. Tung oil, extracted from the fruit of the tung tree, contains large amounts of unsaturated fatty acids. It is a very good biomass fuel oil and excellent drying plant oil, widely used in paints, plastics, artificial rubber, inks, lubricants and biodiesel [1]. Tung trees are deciduous arbors that grow up to 10–20 m in height and are usually dioecious or hermaphroditic with glabrous branches and conical inflorescence consisting of cymes.

The *Vernicia* trees are distributed in eastern Asia and among the three *Vernicia* species, *Vernicia fordii* and *Vernicia montana* are the two primary cultivated plants. As the seed of *V. fordii* displays higher yield and better quality, it has become the main raw material of tung oil industry. However, the *V. fordii* is susceptible to the *Fusarium* wilt disease, which has caused greatly adverse effects on its growth and yield. Fortunately, the *V. montana* was found to be extremely resistant to *Fusarium* wilt disease despite its relatively lower oil production and quality [2]. To date, the disease cannot be managed unless the tree is grafted with *V. montana* (using *V. montana* as the parental stock) [3]. Accordingly, the *V. montana* has become a crucial material for the study of plant disease resistance.

Plants often suffer from many kinds of diseases during their growth and development. In particular, many fungal diseases cause deadly damage to plants. Some plant species have high resistance to various pathogens. These plants are able to recognize and respond to pathogens by activating a variety of defence systems, and the key defence response is the regulation of signaling pathways [4]. In addition, transcription factors (TFs) play a critical role in different signaling pathways, especially in activating pathogen defence signals or inhibiting the expression of downstream defence gene and regulating cross-talk [5,6]. In many instances, genes encoding TFs involved in the resistance to pathogens in *Arabidopsis* have been associated with the ethylene and jasmonic acid signaling pathways [7]. Further, many TF genes families are involved in plant disease resistance, such as MYB genes (MYBs) [8], APETALA2/ethylene response factors (AP2/ERFs) [4], WRKYs [9], NACs [10], basic helix-loop-helix TF [11], as well as the recently described whirly domain TFs [12].

The MYB transcription factors are one of the largest and functionally diverse protein families in the plant kingdom, and they exist in all eukaryotes [13]. The sequences of MYB family are characterized by conserved domain. MYB domain generally consists of three imperfect structural repeats (R1, R2, R3) of 51–52 amino acids, each forming three α–helices [14]. In each repeat of MYBs, the second and third helices build a helix-turn-helix (HTH) structure with three regularly spaced tryptophan residues [15]. Generally, the first tryptophan of R3 is substituted by phenylalanine or isoleucine in plants. MYB proteins have two distinct regions, N-terminal and C-terminal. The conserved DNA-binding domain is located at the N-terminus, while C-terminus is a modulator region responsible for the regulatory activity of the protein [15].

Most MYB proteins function as transcription factors with varying numbers of MYB domain repeats conferring their ability to bind DNA. [15]. Based on the three prototypic MYB protein repeats (R1, R2 and R3), the MYB superfamily can be divided into four subfamilies MYB-related proteins (1R-MYB), R2R3-MYB proteins (2R-MYB), R1R2R3-MYB proteins (3R-MYB) and 4R-like MYB protein (4R-MYB) [16]. The smallest subfamily is the 4R-MYB proteins. They are encoded in several plant genomes, and little is known about their functions [17]. In contrast, the R2R3-MYB proteins are the largest subfamily of MYB proteins in plants, and their functions comprise regulation of secondary metabolism, control of cellular morphogenesis and regulation of meristem formation and the cell cycle [18]. The most diverse subfamily is the MYB-related proteins with a single or a partial MYB repeat, as these can be subdivided into several subclasses [17].

Members of MYB family function in a variety of plant-specific processes. The MYB families are key factors in controlling development, metabolism [17], regulating secondary wall biosynthesis [19,20], and responding to abiotic stresses [21]. The expression of MYB proteins, such as *MYB103*, *MYB85*, *MYB52*, *MYB54*, *MYB69*, *MYB42*, *MYB43*, *MYB20*, has been associated with cells undergoing secondary wall thickening [22]. Overexpression of *MYB46* leads to the activation of secondary wall biosynthetic genes and two secondary wall-associated transcription factors [20]. Recently, the *PtoMYB92* gene was identified as a nuclear-localized transcriptional activator involved in the regulation of secondary cell wall biosynthesis in xylem tissue in *Populus tomentosa* [23].

The comprehensive description and classification of the MYB genes in plant began with the publication of the *Arabidopsis* genome sequence. The functions of MYB genes were further investigated in different plant species, such as maize (*Zea mays*), rice (*Oryza sativa*) [24], petunia (*Petunia hybrida*), snapdragon (*Antirrhinum majus*), grapevine (*Vitis vinifera* L.), poplar (*Populus tremuloides*) and apple

(*Malus domestica*) [17]. Most of the comprehensive studies on MYB superfamily have been carried out on model plants, but very little on *Vernicia*. In this study, we screened the transcriptome sequences of *V. fordii* and *V. montana* to identify MYB genes (*VfMYB* and *VmMYB*). Subsequently, the sequence alignment was applied for the construction of a phylogenetic tree of the two *Vernicia* varieties and *Arabidopsis*. Based on the phylogenetic tree, evolutionary relationships and the potential functions of MYB proteins in *Vernicia* were analysed. The functions of MYB transcription factors associated with resistance and defence against *Fusarium* wilt disease were further speculated by comparing their expression patterns in *V. fordii* and *V. montana* after *Fof*-1 infection.

## 2. Materials and Methods

### 2.1. Plant and Pathogen Materials

In this study, the *V. fordii* and *V. montana* were used as the plant materials. The *V. fordii* and *V. montana* are two species of *Vernicia.* The *V. fordii* seedlings we used were Duiniantong species, which was high yield but susceptible to *Fusarium* disease. The *V. montana* was Guizhou provenance. Seeding plantlets of *V. fordii* and *V. montana*, with 4–6 young leaves, were planted with soil cultivation in greenhouse at 26 °C with a 16 h light/8 h dark cycle, cultured in 5–6 months. The roots, stems, leaves, flower buds, ovaries and kernels tissues were immediately frozen in liquid nitrogen and stored at −80 °C until use. Three independent biological replicates were carried out. The Fof-1 was isolated from the roots of *V. fordii* with wilt disease. The Potato Dextrose Agar (PDA) medium was used for further purification and culture.

### 2.2. Pathogen Inoculation

The pathogen *Fof-1* was isolated from *Fusarium* wilt *V. fordii* in Tianlin County, Guangxi Zhuang Autonomous Region, China. The seeding plantlets of susceptible *V. fordii* and resistant *V. montana* with two or three leaves and a healthy root system were selected for pathogen inoculation. The sterile taproot and lateral roots of the chosen plantlets were longitudinally scratched with a needle and were then dipped in the *Fof-1* spore suspension of $1 \times 10^6$ per mL for 30 minutes. All infected plantlets were replanted in the growth room with 95% relative humidity. According to the symptoms of wilt disease caused by *Fof-1*, the entire root system was harvested at 0, 2, 8 or 13 days after infection and frozen immediately in liquid nitrogen for total RNA extracting. The scratched roots harvested from the uninfected plants at day 0 were the control. Based on the nodes at the sampling times above, the samples were marked as F0, F1, F2, F3 for *V. fordii* and M0, M1, M2, M3 for *V. montana* [3].

### 2.3. RNA Extraction and Expression Analysis

Using the RN38-EASY Spin Plus Plant RNA kit (Aidlab Biotech, Beijing, China), total RNA was extracted from each sample of *V. fordii* and *V. montana* following the manufacturer's protocol. After that, the integrity and the concentrations of the RNA samples were inspected by an Agilent 2100 Bioanalyzer (Agilent Technologies, Santa Clara, CA, USA) and a NanoDrop 5000 spectrophotometer (Thermo Scientific, Waltham, MA, USA). All RNA samples displayed a minimum RNA Integrity Number (RIN) of 7.0 and 260/280 ratio of 2.0. Finally, 1 µg of total RNA was reverse-transcribed for the synthesis of first-strand of cDNA with the superscript III reverse transcriptase (Invitrogen, Grand Island, NY, USA).

### 2.4. Selection and Identification of MYB Sequences in V. fordii and V. montana

The MYB transcription factor family genes were searched from the transcriptome data of the two *Vernicia* species [3] based on the gene annotations. Subsequently, the corresponding ORF and amino acid sequences of the obtained MYB genes in *V. fordii and V. montana* were identified using the NCBI ORF finder (https://www.ncbi.nlm.nih.gov/orffinder/). The presence of the characterized MYB-DNA-binding domain in all obtained protein sequences were confirmed based on

the NCBI Conserved Domain Database (http://www.ncbi.nlm.nih.gov/Structure/cdd/wrpsb.cgi). The sequences without complete MYB DNA-binding domains were eliminated.

## 2.5. Multiple Sequence Alignment and Phylogenetic Analysis

To understand the classification and evolutionary relationship of MYB gene families in *Vernicia*, the MYB genes of *A. thaliana* were download from the *Arabidopsis* Information Resource (https://www.arabidopsis.org/). Multiple sequence alignments were performed on *VmMYBs, VfMYBs and AtMYBs* using the amino acid sequences of the MYB DNA-binding domains in BioEdit 5.0.6 [25]. Subsequently, the phylogenetic trees were constructed using MEGA 7.0.7 (https://www.megasoftware.net/download_form) [26] with the neighbour-joining method, pairwise deletion gap data treatment and a bootstrap of 1000 replicates.

## 2.6. Conserved Motifs Analyses of the MYB Family

To investigate the conserved motifs in the MYB protein sequences, all the full-length protein sequences of *VfMYBs* and *VmMYBs* were analysed by the Multiple Expectation Maximization for Motif Elicitation (MEME) Suite version 4.10.0 tool (http://meme-suite.org/tools/meme) [27]. The parameter settings were used as following: distribution of motifs, zero or one per sequence; maximum number of motifs to find, 20; minimum width of motif, 6; maximum width of motif, 300 (to identify long R2R3 domains). Only motifs with an e-value of $< 1\mathrm{e}^{-20}$ were retained for further analysis.

## 2.7. Evolution and Selection Analysis of the MYB Family

An RBH method was used to identify the one-to-one orthologous genes of putative *MYB* genes between *V. fordii* and *V. montana*. Synonymous (Ks) and non-synonymous (Ka) nucleotide substitution rates were calculated by DnaSP v5.0 software (http://www.ub.edu/dnasp/) [28]. In addition, the Ka/Ks ratios of all of the encoding sites of the *VfMYB* and *VmMYB* paralogs were determined using bio-pipeline-master software (https://github.com/tanghaibao/bio-pipeline/tree/master/synonymous_calculation) [29].

## 2.8. Expression Patterns of MYB Family during Four Infection Processes in V. fordii and V. montana

To explore MYB genes related to disease resistance in *V. fordii* and *V. montana*, expression patterns of each MYB gene were analysed during four infection processes. Reads per kilobase per million mapped reads (RPKM) values were calculated with RSEM software [30] and used to estimate the expression levels of MYB genes. The Multi Experiment Viewer (MeV4) software (http://www.mybiosoftware.com/mev-4-6-2-multiple-experiment-viewer.html) [31] was used to structure a heat map by using the log2-transformed RPKM values compared with the control (0 d).

## 2.9. Network Analysis of VmMYB Family Genes Responding to Fof-1

To explore the hub MYBs highly connected with other genes, the WGCNA was used to construct the co-expressed gene network with a weighted cut-off value >0.50. The co-expression network between MYB hub genes and their connective genes was constructed using Cytoscape software (https://cytoscape.org/download.html) [32]. The functional annotation and classification of the interactive genes were analysed depending on the orthologous groups (COG).

## 2.10. Quantitative Real-Time PCR (RT-qPCR)

According to the non-conserved regions of MYB genes, the specific primers were designed using Primer Premier 5.0. The RT-qPCR analysis was performed with the SYBR® Premix Ex Taq TM Kit (TaKaRa, Tokyo, Japan) and ABI7300 Real-Time quantitative instrument (Applied Biosystems, Foster, CA, USA) in a 20-μl reaction volume. The constitutive gene Actin was used as an internal control. Each sample containing three biological repeats was implemented in technical triplicates. The amplification

conditions were set as 95 °C for 30 s, followed by 40 cycles at 95 °C for 5 s and annealing at 60 °C for 31 s. The amplifications specificity of each target gene was analysed relying on the melting curve, and the relative expression levels of selected genes was calculated using the $2^{-\triangle\triangle Ct}$ method.

## 3. Results

### 3.1. Multiple Sequence Alignment and Phylogenetic Analysis in AtMYBs, VfMYBs, VmMYBs

To explore the MYB transcription factors involved in *Fusarium* wilt resistance in *Vernicia* species, we respectively identified 171 and 177 candidate genes of the MYB superfamily in *V. fordii* and *V. montana* based on the annotations of RNA-seq data. Corresponding ORF analysis revealed that all these obtained genes had one or several conserved domains, which included total or partial MYB-DNA-binding domains. To ensure the accuracy of the sequence, the genes with fragmentary domain were eliminated. Finally, a total of 115 *V. fordii* genes (*VfMYBs*) and 125 *V. montana* genes (*VmMYBs*) were retained. We then used spaces to fill out the missing genes and retained only the conserved domains sequences in the same-length for multiple sequence alignment (Figure S1). Since the MYB sequences in *V. fordii* and *V. montana* had high diversity, some sequences with no common sites were deleted. Finally, a total of 126 proteins from *Arabidopsis*, 75 from *V. fordii*, and 77 from *V. montana* were used for the multiple sequence alignment and the construction of phylogenetic tree. In the multiple sequence alignment, the conservative amino acids MYB domains of *V. fordii* and *V. montana* were similar to *Arabidopsis*, in which the primary structure of R2 and R3 DNA-binding domains is [-W-(X19)-W-(X19)-W- . . . . . . -F/I-(X18)-W-(X18)-W-] (W represents tryptophan and X represents any amino acid) [17].

According to the full-length alignments of *VfR2R3-MYBs* and *VmR2R3-MYBs*, the sequence logos of R2 and R3 repeats were analysed. The height of each letter indicates the conservation degree of the amino acid residue at the corresponding site, and the asterisks indicate conserved tryptophan (Trp) residues in the MYB domain (Figure 1). There were three extremely conserved Trp residues with red asterisks between the approximately 54 amino acids in the R2 motif in Figure 1a, and the first Trp site in the R3 motif with a green asterisk was frequently replaced by other amino acids (Figure 1b).

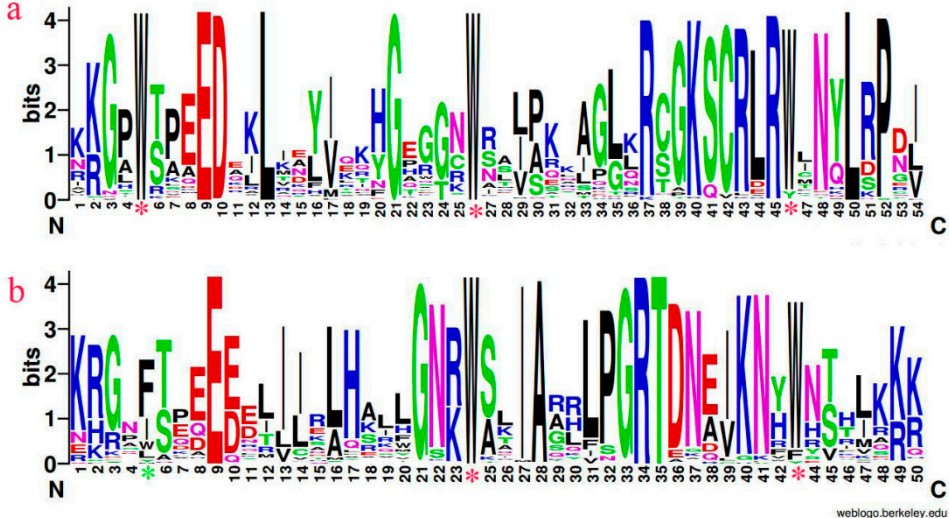

**Figure 1.** The sequence logos of R2 (**a**) and R3 (**b**) repeats are distributed in all R2R3-MYB proteins of the *Vernicia* species. The analysis of conserved domains of the R2 and R3 MYB sequences are based on all the full-length alignments of *VfR2R3-MYB* and *VmR2R3-MYB* proteins in MEME Suite version 4.10.0.

Based on the multiple sequence alignment, 126 *AtMYBs*, 75 *VfMYBs* and 77 *VmMYBs* were used for the construction of the UPGMA (unweighted pair-group method with arithmetic means) phylogenetic tree with MEGA 7.0.7 software (Figure 2). As previous reported, the AtR2R3-MYB proteins had been

divided into sub-groups on the basis of the conserved DNA-binding domain and amino acid motifs in the C-terminal domains, which were marked as S1–S25 in our supported evolutionary tree [17] (Figure 2). These conserved motifs facilitate to identify the functional domains outside of the MYB DNA-binding domain [33]. In the supported phylogenetic tree, all MYB proteins were divided into 35 primary groups, which named C1 to C35 and distinguished by different colors. As shown in Figure 2, several subgroups were identified in *Vernicia* for which there were no representatives in *Arabidopsis*. In detail, *VfMYBs* and *VmMYBs* without *AtMYBs* existed in the C16, C19, C28, C29, C32, C33 and C35 subgroups.

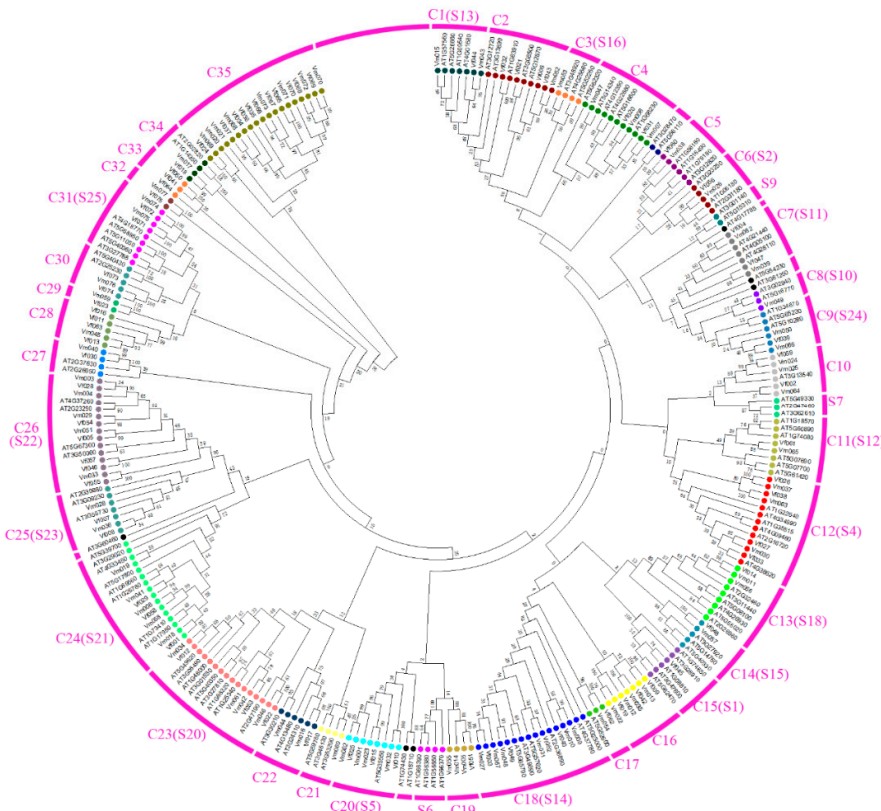

**Figure 2.** Phylogenetic relationships of MYB family proteins in *Arabidopsis*, *V. fordii* and *V. montana*. 278 MYB proteins from *Arabidopsis*, *V. fordii* and *V. montana* were used to do the multiple sequence alignment and construct the unrooted phylogenetic tree. BioEdit 5.0.6 with the default settings and MEGA 7.0.7 with the UPGMA method and 1000 bootstrap replications were used. Different colored dots represent different subgroups.

### 3.2. Distribution of Conserved Motifs Outside of the MYBs Domain

The regions outside the DNA-binding domain in transcription factors are often conserved and genes with these motifs are likely to have similar functions [1]. To identify the potential MYB genes with similar functions in *V. fordii* and *V. montana*, 64 *VfMYBs* and 69 *VmMYBs* were used for the identification of conserved motifs by the MEME online tool. An unrooted phylogenetic tree in Figure 3 was constructed using MEGA 7.0.7 software with the neighbour-joining (NJ) method and 1000 bootstrap replicates. It was similar to the phylogenetic tree created with the UPGMA method to a great extent, which revealed that the phylogenetic relationships and classification of the abovementioned MYB family proteins were highly reliable. Of the twenty motifs, Motifs 1, 3, 4 and 18 were present in many MYBs, especially Motif 1, which was present in more than fifty percent of the MYBs (Figure 3). The results indicated that the great majority of members in the same subfamily showed similar motif distributions. However, some motifs only existed in specific subgroups. It could represent the particular function of the corresponding subgroup. As shown in Figure 3, Motif 14 was only found in C26, whereas the members of *Vf046* and *Vf057* had Motif 5 instead of Motif 14. In addition, Motif 10

only existed in members of C25. Specifically, Motif 15 was only found at the N-terminal region of C25 in *V. fordii*, and it could be considered as a marker to distinguish the two species of *Vernicia*. Similarly, Motif 12 and Motif 20 were unique to the C23 and C34 subgroups, respectively.

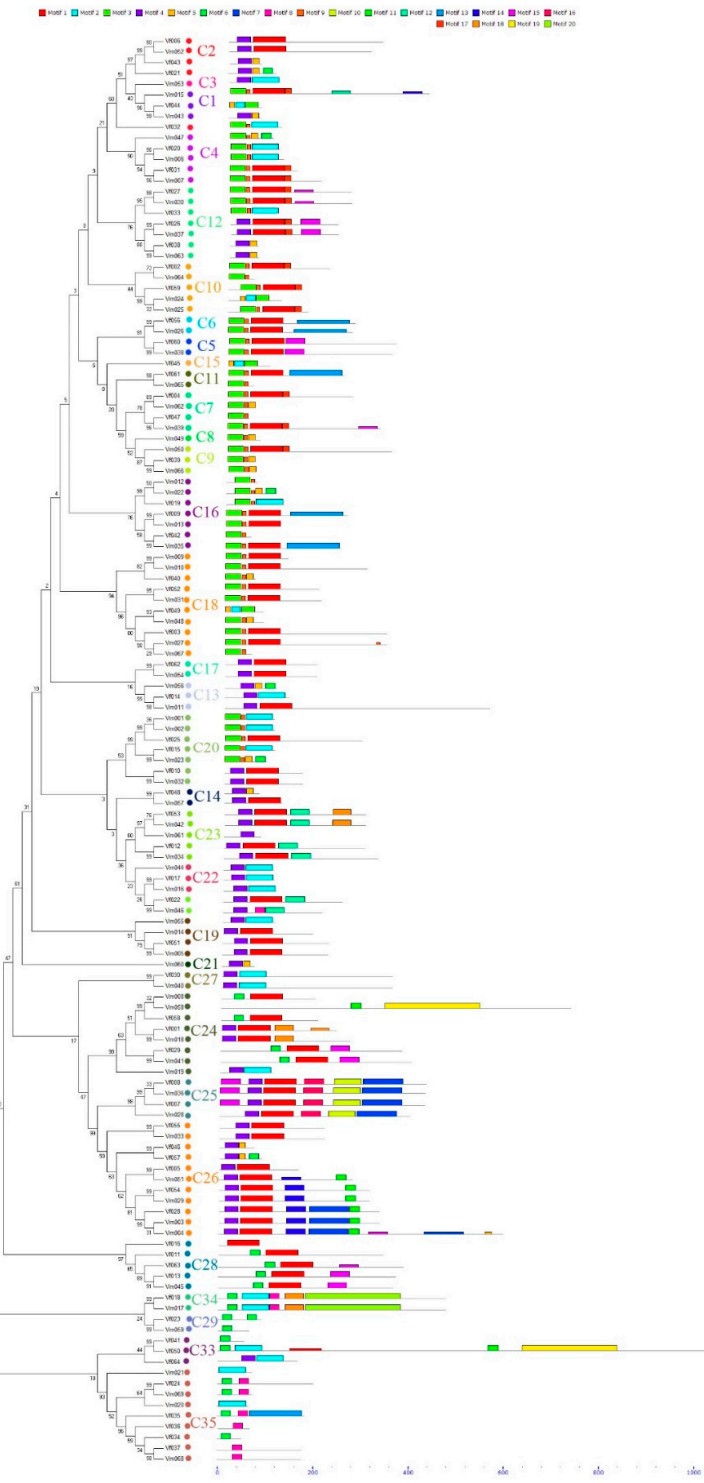

**Figure 3.** The conserved motifs of MYBs in *V. fordii* and *V. montana*. The unrooted phylogenetic tree of 64 *VfMYBs* and 69 *VmMYBs* was constructed by MEGA 7.0.7 software with the neighbour-joining (NJ) method and 1000 bootstrap replications. All subgroups named C1 to C35 are marked in different colors. The analysis of conserved motifs was carried out using the MEME online tool, which was set to search for at most twenty motifs with lengths between 6 and 200 amino acids. The colored boxes represent corresponding motifs.

### 3.3. The Analysis of Evolution and Selection in MYB Family by Ka/Ks Ratios Calculation

To clearly reveal the evolutionary mechanism of MYB family in *Vernicia*, a total of 49 pairs of potential one-to-one orthologous genes were successively screened between *V. fordii* and *V. montana* (Table S1). In addition, the nonsynonymous and synonymous substitution ratio (Ka/Ks) of the duplicated genes was calculated (Table S1) to identify the evolutionary forces of MYB genes. The ratio of Ka to Ks was divided into three cases, which represent different meanings. Ka/Ks = 1 was chosen as the cut-off to identify genes under positive selection or not [34]. Normally, Ka/Ks < 1 indicates negative or purifying selection, Ka/Ks > 1 indicates positive or adaptive selection, and Ka/Ks = 1 indicates neutral evolution. In this study, 22 pairs of orthologous genes between *V. fordii* and *V. montana* definitely had the same sequence, with a Ka/ks value of zero. In terms of the Ka/Ks ratios of other orthologs, 81.5% (22/27) of orthologous genes showed Ka/Ks ratio < 1, while 18.5% (5/27) exhibited Ka/Ks > 1. As calculated in Table S1, a significant proportion of MYB orthologous genes were undergone purifying selection, while a small proportion of MYBs were undergone adaptive selection.

### 3.4. Expression Patterns of MYB Family in V. Fordii and V. montana during Infection

To further explore the expression levels of MYB genes under *Fof-1* infection, we performed the expression pattern clustering analysis at four infection phases (0, 1, 2, 3) in *V. fordii* and *V. montana* (Figures S2 and S3). The MeV4 software package was used to create a heat map with the normalized RPKM values. The color strip at the top indicates log2-transformed RPKM values. Both upregulated and downregulated expressions were reflected in MYB genes in the two species of *Vernicia* (Figures S2 and S3). In *V. fordii*, twenty-two genes (29.3%) showed downregulated expression among the three stages, while 12 genes (16%) were upregulated (Figure S2). In *V. montana*, 27 (35.1%) MYBs were downregulated and 17 (22.1%) were upregulated during pathogen infection (Figure S3).

To further investigate the expression correlation of the MYB genes between *V. fordii* and *V. montana*, the expression profiles of the 49 pairs of orthologous genes during *Fof-1* infection were studied (Figure 4). Quite a number of orthologous genes exhibited similar expression patterns, in which 18 (36.7%) pairs of orthologous MYBs showed upregulation in both *Vernicia* species. In contrast, a few of orthologous MYBs exhibited opposite expression trends. It is worth noting that four pairs of orthologous genes (*VfMYB003/VmMYB027*, *VfMYB053/VmMYB042*, *VfMYB050/VmMYB058*, *VfMYB054/VmMYB029*) marked in red background, at the bottom of Figure 4, showed upregulation in *V. fordii* but downregulation in *V. montana* during the infection process.

In addition, a large proportion of orthologous genes had fluctuating expression patterns between upregulation and downregulation. Generally, 14 pairs orthologous MYBs showed upregulated expression in the first phase and downregulated expression in the second phase, but showed a slightly different tendency in the third and fourth stages. The *VfMYB017* marked in green background was downregulated during the last two phases, while *VmMYB016* was slightly downregulated in the third stage and upregulated in the last stage. *VfMYB039* showed increased expression in the third phase and decreased expression in the last phase, and the expression pattern of *VfMYB066* was contrary to that of *VfMYB039*.

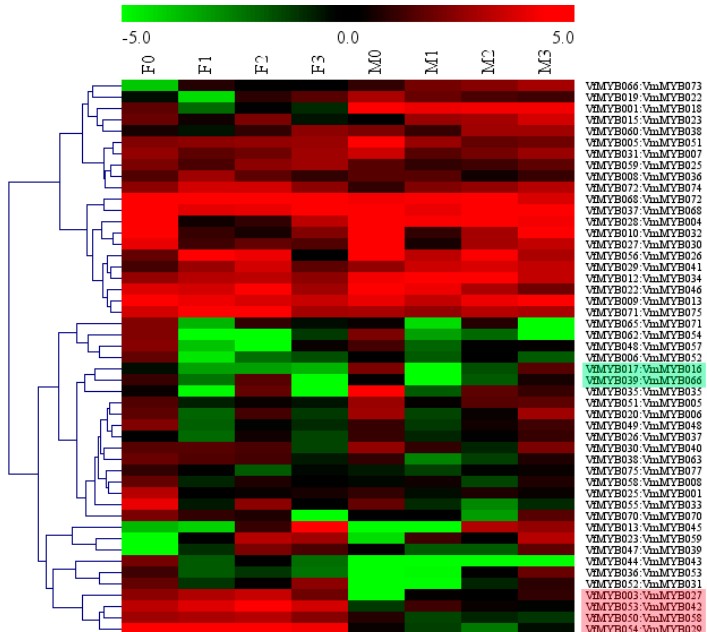

**Figure 4.** Expression profiles of Vf/VmMYB homologous genes under infection by *Fof-1* in *V. fordii* and *V. montana*. F0-F3 represented the expression of VfMYB homologous genes in *V. fordii* during the infection stage (0, 1, 2, 3) by the pathogen *Fof* 1; M0-M3 represented the expression of VmMYB homologous genes in *V. montana* during the infection stage (0, 1, 2, 3) by the pathogen *Fof* 1. The expression levels were illustrated by the gradient color. Red and green represent high and low expression level respectively.

## 3.5. Network Analysis Based on Four Hub Genes Affording Resistance to Fof-1 in V. montana

To identify the interaction between MYB family genes and other genes in *V. montana*, a WGCNA was used to determine hub genes that highly connected with other genes. On the basis of a weighted cut-off value >0.50, four hub genes (*VmMYB013*, *VmMYB034*, *VmMYB011*, *VmMYB041*) showing extremely high interaction with the other 640 genes were identified. Based on the COG functional annotations, 634 genes with biological function were divided into 21 different clusters, other six genes with unknown function were rejected. Of the 634 annotated genes, 91 genes (14.4%) clustered into post-translational modification, protein turnover, chaperones, followed by those clustered into signal transduction mechanisms (79; 12.5%) and general function prediction only (71; 11.2%). In general, some genes directly or indirectly responded to plant disease resistance, such as those clustered into RNA processing and modification, amino acid transport and metabolism, carbohydrate transport and metabolism, lipid transport and metabolism and transcription. However, a small number of genes, classified into defence mechanisms, cell wall/membrane/envelope biogenesis and secondary metabolites biosynthesis, transport and catabolism, were not to be neglected. Most of the marginal genes interacted with one or two hub genes, while genes in the core region connected with three or four hub genes (Figure 5).

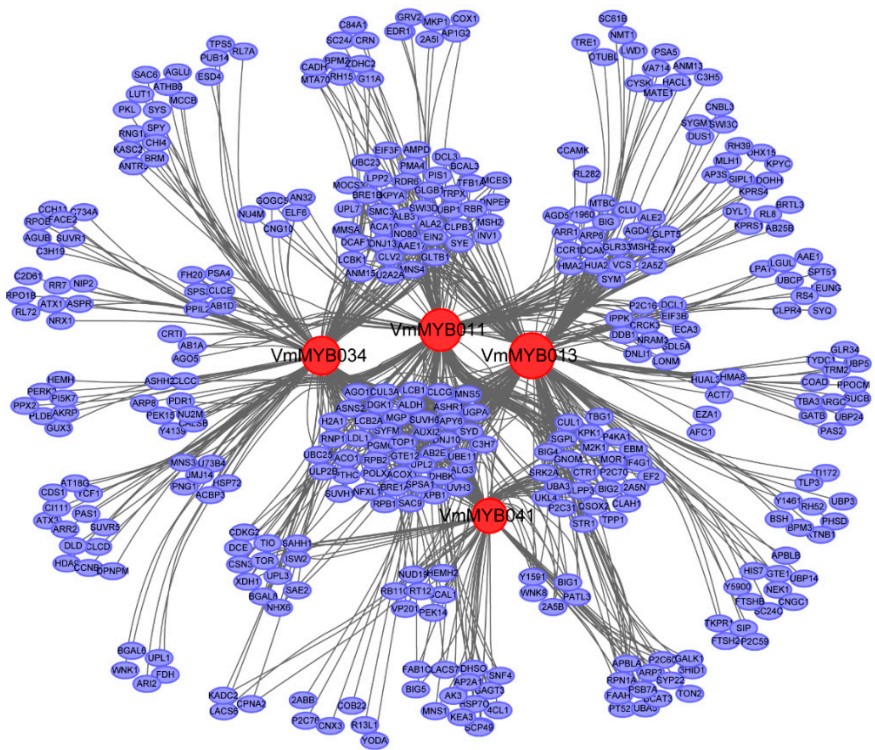

**Figure 5.** Network analysis of *VmMYB013, VmMYB034, VmMYB011* and *VmMYB041* hub genes in response to *Fof-1*. The four hub genes interacted with 634 genes and exhibited a total of 1157 interactions. The red nodes represent the four hub genes. Bottle blue nodes represent the interaction genes.

### 3.6. Tissue-Specific Expression Pattern of Differentially Expressed MYB Genes in Response to Fof-1

To further excavate the key genes related to pathogen resistance, the expression patterns of three hub *VmMYB* genes and their corresponding orthologous genes were performed. Root, stem, leaf and kernel tissues as well as stamen, petal and bud tissues in *V. fordii* and *V. montana* were used for the tissue-specific expression analysis of *VmMYB011/VfMYB011*, *VmMYB013/VfMYB009* and *VmMYB041/VfMYB029* (Figure 6a). Meanwhile, the expression patterns of these three hub genes in root tissue during the four infection periods were validated (Figure 6b). As shown in Figure 6a, both *VmMYB011/VfMYB011* and *VmMYB041/VfMYB029* showed the highest expression levels in kernel, while *VmMYB013/VfMYB009* were the highest-expressed genes in root. It indicated that they may function mainly in kernel and root, respectively. The expression of *VmMYB041* in root was five times higher than that of *VfMYB029* (a significant difference) (Figure 6a). In addition, *VmMYB041* and the corresponding orthologs *VfMYB029* showed completely opposite expression patterns in root tissue during the four infection periods (Figure 6b). *VmMYB041* demonstrated the trend of rising first and then decreasing, which began the transition in second stages and then declined. In contrast, *VfMYB029* showed a trend of decreasing in the first infection period and then increasing. The high expression level in root and the induced expression at the first two infection stages of *VmMYB041* suggests that it may play an important function in disease resistance in root. Although the expression of *VmMYB011/VfMYB011* was not the highest in root tissue, their expression patterns were different from each other, which suggested their functional divergence in response to wilt disease. In the first infection period, the expression level of *VmMYB011* reached the peak, while the expression of *VfMYB011* reached the lowest level.

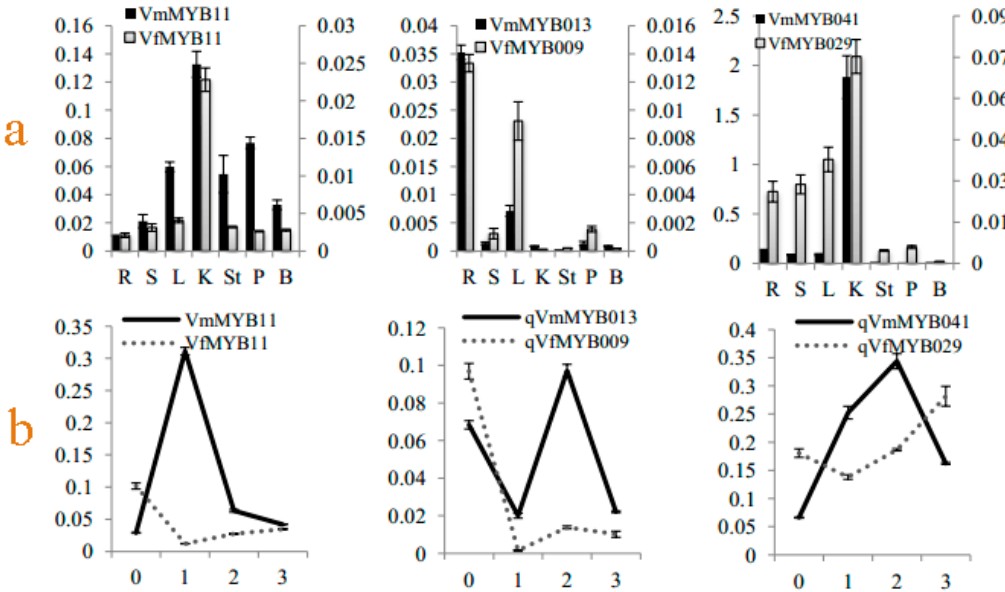

**Figure 6.** Tissue-specific expression analyses of three hub genes in *Vernicia*. The y-axis indicates the relative levels of expression. The left vertical axis indicates the expression levels of *VmMYB*; the right vertical axis indicates the expression levels of *VfMYB*. The expression levels were analysed using RT-qPCR in triplicate. The x-axis indicates root (R), stem (S), leaf (L), kernel (K), stamen (St), petal (P) and bud (B) tissues used for the expression analysis (**a**). Root tissues of *V. fordii* and *V. montana* were used for the expression analysis responding to *Fof-1* (**b**). The numbers in the x-axis represent the four stages of infection, as follows: 0, uninfected stage; 1, early stage of infection; 2, middle stage of infection; 3, late stage of infection.

## 4. Discussion

### 4.1. The R2R3 MYB Subfamily Plays a Crucial Role in Vernicia and the Systematic Evolution Gradually Tend to Diversity

Due to the high diversity of the MYB family in the process of multiple sequence alignment, it had no common loci in some sequences. Therefore, 75 *VfMYBs* and 77 *VmMYBs* were chosen for the analysis of the MYB gene family. The numbers of MYBs in *Vernicia* were less than that in *Arabidopsis thaliana* and physic nut (Table 1). In *V. fordii* and *V. montana*, 64 and 69 genes were assigned to the R2R3-MYBs, and only one gene was assigned to the 4R-MYB subfamily, respectively. It is similar to the classification of *A. thaliana* and physic nut. As previous studies have found, these findings illustrate that the R2R3 MYB subfamily occupied a large proportion of the whole MYB family. According to the unrooted phylogenetic tree (Figure 2), most subgroups in *Arabidopsis* were identified in the two species of *Vernicia*. As previous reports, the subgroup S2 of *Arabidopsis* was proved to control the biosynthesis of proanthocyanidins (PAs) in the seed coat [17] and subgroup S4 was proved to encode transcriptional repressors [35–37]. In addition, the subgroup S21 has been shown to actively regulate the thickening of the cell walls in fiber cells and subgroup S13 has been influenced lignin deposition, and mucilage production [17]. However, most subgroups of these MYB proteins, defined first in *Arabidopsis*, are also present or sometimes expanded in other plants [38]. It was confirmed in this study that several subgroups exiting in *V. fordii* and *V. montana* but not in *Arabidopsis*, which suggested that these proteins might have specialized roles or were acquired after divergence from the last common ancestor [17]. This illustrated that some MYB genes tended to diverge, gradually, in the systematic evolution of the different species.

**Table 1.** Numbers of the MYB family in *V. montana*, *V. fordii*, *Arabidopsis thaliana* and *Jatropha curcas*.

| MYB Protein Classes | *V. fordii* | *V. montana* | *A. thaliana* | *J. curcas* |
|:---:|:---:|:---:|:---:|:---:|
| R2R3 | 64 | 69 | 126 | 123 |
| 1R | 6 | 4 | 64 | n.d. |
| 3R | 4 | 3 | 5 | 4 |
| 4R | 1 | 1 | 1 | 1 |
| Total | 75 | 77 | 196 | 128 |

Annotations: The abbreviation n.d. means not determined and the dates of *Arabidopsis thaliana* and *J. curcas* reference to previous studies [12,39].

### 4.2. Four Pairs of Vf/VmMYBs Had Negatively Correlated Expression between Two Species in Response to Fof-1

As shown in Figure 4, most orthologous genes exhibited similar expression tendencies in response to wilt disease, and only several genes showed relatively divergent expression patterns. At the bottom of Figure 4, the four pairs of orthologous genes, *VfMYB003/VmMYB027, VfMYB053/VmMYB042, VfMYB050/VmMYB058*, and *VfMYB054/VmMYB029-*, were upregulated in *V. fordii* but downregulated in *V. montana* during the infection process. The *Arabidopsis thaliana* gene *AtMYB62*, the homologue of *VfMYB053/VmMYB042*, is located in the nucleus, and its overexpression results in altered root architecture, phosphate (Pi) uptake, and acid phosphatase activity, leading to the decrease of total Pi content in the shoots [40]. Meanwhile, the overexpression of *AtMYB62* results in a characteristic gibberellic acid (GA)-deficient phenotype that can be partially reversed by exogenous application of GA [40]. Many of the morphological changes observed during Pi stress, such as altered root system architecture and root/shoot ratio, are coordinated by phytohormones such as auxin, ethylene, and cytokinin [41]. In addition, the effective local variation of phosphorus will affect the number of lateral root and root hair [41]. Root hair length decreases logarithmically in response to increasing phosphorus concentration [42]. As shown in Figure 4, *VfMYB053* and *VmMYB042* showed opposite expression patterns in the two *Vernicia* species responding to *Fof-1*. Interestingly, the growth rate of lateral roots and root hairs is extremely fast in *V. montana* but very slow in *V. fordii* after *Fof-1* infection.

*VmMYB003* was downregulated in *V. montana* during the four *Fof-1* infection phases (Figure S3). *AtMYB73* in *Arabidopsis* was identified as a homologous gene of *VmMYB003*. The *AtMYB73* gene has an important role in resistance to *Sclerotinia sclerotiorum* in *Arabidopsis thaliana* through the SA signal pathway and JA signal pathway. Similarly, *MYB73* is involved in the NPR1-mediated SA and JA signaling pathways of *Bipolaris oryzae* [43].

### 4.3. Functional Speculation of Four Hub Genes in Resistant V. montana

We performed functional analysis of the homologues of these four hub genes in *Arabidopsis*. *AtMYB33*, the homologous gene of *VmMYB011* in *Arabidopsis*, is a positive regulators of abscisic acid (ABA) signaling. The expression analysis of pre-miR159 and *AtMYB33* reveals a possible link between CBP80 (ABH1) and ABI4 [44]. The plant-stress hormone abscisic acid (ABA) has been implicated in plant defence responses [45]. The ABA-dependent signaling pathway regulates stress-inducible gene expression through several positive and negative regulators and the ABA-signaling mutant *aba2-1*, shows increased resistance to the necrotrophic fungal pathogen *F. oxysporum* in *Arabidopsis* [6]. The homologous gene of *VmMYB013* in rice directly upregulates the expression of a secondary wall-specific cellulose synthase gene, cellulose synthase A7 (OsCesA7) [46]. This gene's homologue in *Arabidopsis thaliana* is a transcriptional regulator specifically activating lignin biosynthetic genes during secondary wall formation [47]. In addition, *GhMYB108*, the homologous gene of *VmMYB034* in cotton, forms a positive-feedback loop with *CML11* and participates in the defence response against *Verticillium dahliae* infection [48]. GhMYB108 interacts with calmodulin protein GhCML11 to form a positive-feedback loop to enhance the transcription of *GhCML11* in a calcium-dependent manner. Influx of $Ca^{2+}$ into the cytosol as an early event plays an important role in pathogen attack [49].

The homologous gene of *VfMYB041* in *Arabidopsis thaliana*, *AtMYB56*, is a novel target of a CULLIN3 (CUL3)-based E3 ligase and a negative regulator of flowering by controlling the expression of the FLOWERING LOCUS T (FT) [50].

*4.4. Tissue-Specific Expression Pattern of Different MYB Homologous Genes in Response to Fof-1*

Two pairs of orthologous genes, *VmMYB011/VfMYB011* and *VmMYB041/VfMYB029*, showed opposite expression patterns between *V. montana* and *V. fordii* (Figure 6). The expression levels of *VmMYB011* and *VmMYB041* showed a rapid increase at the first infection period in *V. montana*. In addition, the expression levels of these two genes in root tissue in *V. montana* was several times as high as their orthologous genes in *V. fordii* (Figure 6a). The homologous gene of *VmMYB011* in *Arabidopsis* positively regulates abscisic acid (ABA) signaling which is implicated in plant defence responses [42]. Similarly, the *VmMYB041* homologue marks specific proteins for degradation [50].

## 5. Conclusions

This study first reports the identification and analysis of the MYB family in *Vernicia* in response to wilt disease. We explored phylogenetic relationships, conserved motifs, evolutionary selection and expression patterns of MYB genes in response to wilt disease between *V. fordii* and *V. montana*. Based on the comparison of MYBs in the two *Vernicia* species, the function and transcriptional expression of some genes have become clearer. Our findings will be crucial not only for understanding the diversity and functionality of the MYB family but also for determining the mechanism of resistance to *Fusarium* wilt, which will provide a valuable basis for future research on MYBs in *Vernicia*.

**Supplementary Materials:** The following are available online at http://www.mdpi.com/1999-4907/10/2/193/s1, Figure S1: Multiple sequence alignments for the sequence with complete MYB domain. Multiple sequence alignment was accomplished by BioEdit (https://bioedit.software.informer.com/) and DNAMAN (https://www.lynnon.com/pc/framepc.html) software. The blue shadow, pink shadow and black shadow, respectively, represent the proportion of conserved amino acid residues >50%, >75% and 100%; Figure S2: Expression profiles under infection by *F. oxysporum* in *V. fordii*; Figure S3: Expression profiles under infection by *F. oxysporum* in *V. montana*; Table S1: The nonsynonymous and synonymous substitution ratio (Ka and Ks) of the duplicated genes.

**Author Contributions:** Y.C. designed the experiments; X.W. and Q.Z. analysed the data; M.G., L.W. and Y.W. provided technical assistance; X.W. wrote the article with contributions from all the authors; Y.C. and Y.W. conceived the project, supervised the analysis and critically reviewed the manuscript. All authors read and approved the final manuscript.

**Funding:** The work has been financially supported by science and Technology Major Program on Agricultural New Variety Breeding of Zhejiang, China (no. 2016C02056); "Ten thousand people plan" Science and Technology Innovation Leading Talent of Zhejiang, China (no.2018R52006); National Key R&D Program of China (no. 2017YFD0600704) awarded to Y.-C.C.

**Conflicts of Interest:** The authors declare that there are no competing interests.

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
