# Peer review of "Expression Patterns of MYB (V-myb Myeloblastosis Viral Oncogene Homolog) Gene Family in Resistant and Susceptible Tung Trees Responding to Fusarium Wilt Disease"

_forests, doi:10.3390/f10020193_

Round 1

Reviewer 1 Report

The manuscript is written very clearly and results along with discussion is crisp. The MS can be accepted for publication with following minor changes: 1. qRT-PCR should be replaced with RT-qPCR throughout the manuscript.

2. On page 4, line 167: new para should start from point 2.10

3. Authors should discuss in brief significance of presence or absence of MYB sequences from Arabidopsis and Vernicia in different subgroups presented in Fig. 2

4. The authors can indicate pair of orthologous genes they talked about in manuscript through arrow in Fig. 4

5. Authors should discuss the significance of varied expression of hub genes in various tissues as well as reason/significance of varied expression of these genes during various stages of infection as shown in Fig. 6

Author Response

Detailed response to reviewer :

1. qRT-PCR should be replaced with RT-qPCR throughout the manuscript.

Response: We used RT-qPCR instead of qRT PCR throughout the manuscript.

2. On page 4, line 167: new para should start from point 2.10

Response: The new para was start from point 2.10 on page 4, line 173.

3. Authors should discuss in brief significance of presence or absence of MYB sequences from Arabidopsis and Vernicia in different subgroups presented in Fig. 2

Response: A discuss in functions of MYBs from Arabidopsis in different subgroups was supplemented in point 4.1 from “As previous reports, the subgroup S2 of Arabidopsis was…”. As MYB genes of Vernicia have not been investigated and reported, they are not detailed described in this manuscript.

4. The authors can indicate pair of orthologous genes they talked about in manuscript through arrow in Fig. 4

Response: According to the reviewer’s suggestion, we marked the four pairs of orthologous genes  in red background at the bottom of Fig. 4, and the other orthologous genes in green background. Please refer to the page 9, Fig. 4.

5. Authors should discuss the significance of varied expression of hub genes in various tissues as well as reason/significance of varied expression of these genes during various stages of infection as shown in Fig. 6

Response: According to the reviewer’s comments, we discussed the significance of varied expression of hub genes in various tissues in the section “3.6. Tissue-specific expression pattern of differentially expressed MYB genes in response to Fof-1. Please refer to line 322 “It indicated that they may…”, line 328 “The high expression…” and line 332 “which suggesting their functional divergence…”.

Reviewer 2 Report

Wang et al., evaluated expression modes and internal network of MYB transcription factors in two Vernicia tree species, one resistant (V. fordii) and one susceptible (V. montana), in response to Fusarium wilt (Fusarium oxysporum f. sp. fordii 1, Fof-1). The authors found a total of 75 VfMYB and 77 VmMYB genes. In addition, they detected 49 pairs of one-to-one orthologous Vf/VmMYB genes. They then investigated their expression patterns and found that most orthologous Vf/VmMYB genes exhibited similar expression patterns in response to Fof-1 infection. Interestingly, seven pairs of Vf/VmMYB genes were expressed differently across the different tree genotypes. The authors conclude that their study set the basis to understand the diversity and functionality of the MYB transcription factor family but also to determine the genetic basis of resistance to fusarium wilt disease.

Other comments:

* the authors should describe better the plant material that was used. which genotypes were used? are they inbreed? how do they differ genetically? 

* the authors should describe better the plant growing conditions. which substrate was used? which age of the plant was used? 

Author Response

Detailed response to reviewer 2:

1. The authors should describe better the plant material that was used. which genotypes were used? are they inbreed? how do they differ genetically?

Response: In this study, the V. fordii and V. montana were used as the plant materials. The V. fordii seedlings we used were Duiniantong variety, which has high yield production but susceptible ability to Fusarium disease. The V. montana seedling were Guizhou provenance, there were no varieties investigation for V. montana so far. V. montana species owns high resistance to Fusarium disease. Please refer to the part of “2.1. Plant and pathogen materials” in line 103 to106.

2. The authors should describe better the plant growing conditions. which substrate was used? which age of the plant was used?

Response: Seeding plantlets of Vernicia species, V. fordii and V. montana seedlings with 4-6 young leaves were planted with soil cultivation in greenhouse at 26 °C with a 16 h light/8 h dark cycle, cultured in 5-6 months. We added the description of plant growing conditions in the section “2.1. Plant and pathogen materials” in line 106-107.

Reviewer 3 Report

In this manuscript, the authors compared two Vernici species (ferdii and montana) in terms of response to a soil-borne fungus, focusing on the MYB transcription factors to identify which of them might be responsible for the difference in response. Overall, this manuscript tackles an interesting question and gives a detailed and informative introduction, however there are some points, outlined below, that need to be addressed:

The references in some sections are missing. One example is line 18, where a previous study is mentioned but no reference is given.

Please outline the abbreviations in the first use, even though they are in the abstract. I was not sure what RBH stands for until I reached Methods section.

In line 20, please provide how the 75 and 77 MYB transcription factors were identified. Were they identified through differential expression?

In line 25, what does "labeled as unknown proteins" mean? Does this refer to a database?

I found the Methods section to be confusing because the sections are not in order and have missing information. For example after section 2.1 Plant and pathogen material, I was expecting to learn how the plant material was further handled and how the RNA extraction was conducted but that section comes at the end of Methods. In 2.2, transcriptome data is mentioned without any reference. Did the authors generate the transcriptome data? Did it come from a previous study?

In line 150, the authors mention that they calculated RPKM, but no tool is referenced. There are a variety of tools to calculate RPKM. Even if it is a custom tool, it should be mentioned.

In line 151, please provide a reference for the MeV tool.

In the results section 3.1, I was not sure what the comprehensive investigation was to identify the 115 and 125 MYB genes and what the goal was. The overall goal of this study is clear, however in subsections, if the authors could state their sub-goals, that would be informative. For example what is the goal of looking into the distribution of conserved motifs outside the MYB domain in section 3.2?

In the figures, please label each axis and color key in the figure. In some of the figures the information is missing (Figure 4 -color key- what does high/low expression refers to?) or it is given in the figure legend (Figure 6- axis labels).

Overall please avoid using vague terms such as "remarkable (line17)", "comprehensive investigation (line179), "clearly (line237)", "extremely down-regulated (line 360)". In some of these instances, a statistical test result would be more convincing to express the results instead of an adjective.

Author Response

Detailed response to reviewer 3:

1. The references in some sections are missing. One example is line 18, where a previous study is mentioned but no reference is given.

Response: According to the reviewer’s comments, we added the reference of our previous comparative transcriptomic results “[3]” in line 18. And we check the references in the manuscript.

2. Please outline the abbreviations in the first use, even though they are in the abstract. I was not sure what RBH stands for until I reached Methods section.

Response: According to the reviewer’s suggestion, we have strictly checked the usage of abbreviations. For example, the full names of RBH and WGCNA were added in the part of abstract in line 21 and line 28 when they first use.

3. In line 20, please provide how the 75 and 77 MYB transcription factors were identified. Were they identified through differential expression?

Response: According to the reviewer’s suggestion, we supplied how the 75 and 77 MYB transcription factors were identified in line 18 “Depending on whether the sequence has a complete MYB-DNA-binding domain, a total of 75 VfMYB and 77 VmMYB genes were identified in V. fordii and V. montana, respectively”.

4. In line 25, what does "labeled as unknown proteins" mean? Does this refer to a database?

Response: In this study, the "labeled as unknown proteins" means annotated as unknown proteins, which was based on a database of transcriptome functional annotation. We revised it. Please refer to line 27.

5. I found the Methods section to be confusing because the sections are not in order and have missing information. For example after section 2.1 Plant and pathogen material, I was expecting to learn how the plant material was further handled and how the RNA extraction was conducted but that section comes at the end of Methods.

Response: According to the reviewer’s comments, the materials and methods section has been extensively revised. The part of “Pathogen inoculation” was moved from point 2.6 to point 2.2 and the part of “RNA extraction and expression analysis” was moved from point 2.9 to point 2.3, please refer to point 2.2 to 2.10.

6. In 2.2, transcriptome data is mentioned without any reference. Did the authors generate the transcriptome data? Did it come from a previous study?

Response: The transcriptome data of the two Vernicia species from our previous comparative transcriptomic results. According to the reviewer’s suggestion, the reference of “[3]” was supplemented in line 135.

7. In line 150, the authors mention that they calculated RPKM, but no tool is referenced. There are a variety of tools to calculate RPKM. Even if it is a custom tool, it should be mentioned.

Response: We added the description about the calculated tool for RPKM “RSEM software” and the reference “[30]” to the manuscript, please refer to line 164, page4.

8. In line 151, please provide a reference for the MeV tool.

Response: We added the reference “[31]” for the MeV tool, please refer to line 165, page4.

9. In the results section 3.1, I was not sure what the comprehensive investigation was to identify the 115 and 125 MYB genes and what the goal was.

Response: According to the reviewer’s suggestion, the investigation processes and the goal were added to the results section 3.1, please refer to line 360, page 17.

10. The overall goal of this study is clear, however in subsections, if the authors could state their sub-goals, that would be informative. For example what is the goal of looking into the distribution of conserved motifs outside the MYB domain in section 3.2?

Response: According to the reviewer’s suggestion, the goal of section 3.2 was added to the manuscript, please refer to line 227, page 6.

11. In the figures, please label each axis and color key in the figure. In some of the figures the information is missing (Figure 4 -color key- what does high/low expression refers to?) or it is given in the figure legend (Figure 6- axis labels).

Response: According to the reviewer’s suggestion, we checked the Figure information and Figure legends. In the Figure 4, we revised the figure description “Expression profiles of Vf/VmMYB homologous genes under infection by Fof-1 in V. fordii and V. montana. F0-F3 represented the expression of VfMYB homologous genes in V. fordii during the infection stage (0, 1, 2, 3) by the pathogen Fof1; M0-M3 represented the expression of VmMYB homologous genes in V. montana during the infection stage (0, 1, 2, 3) by the pathogen Fof1. The expression levels were illustrated by the gradient colour. Red and green represented high and low expression level, respectively.” And the axis labels in Figure 6 were supplemented, please refer to line 339 and line 341, page 11.

12. Overall please avoid using vague terms such as "remarkable (line17)", "comprehensive investigation (line179), "clearly (line237)", "extremely down-regulated (line 360)". In some of these instances, a statistical test result would be more convincing to express the results instead of an adjective.

Response: According to the reviewer’s suggestion, we deleted the vague terms, such as "remarkable ", "comprehensive’’, "clearly " and "extremely ".

Round 2

Reviewer 3 Report

With the major revisions, the manuscript is clearer. After a minor revision of the following points, the paper could proceed to publication:

-line 371: "obviously upregulation"?

-line 383 Is the heatmap reflecting expression level or the fold change expression? Since the scale goes from negative to positive, please clarify what is being compared in the fold change.

-line 448 "which suggesting"

Author Response

Detailed response to reviewer 1:

1. -line 371: "obviously upregulation"?

Response: According to the reviewer’s comments, we re-checked and modified the use of vague terms, such as we replaced "obviously upregulation" with " upregulation", replaced “different” with “significantly different” and replaced “obviously less than” with “less than”. Please refer to line 279, 334 and 352, page 8, 10 and 11.

2. -line 383 Is the heatmap reflecting expression level or the fold change expression? Since the scale goes from negative to positive, please clarify what is being compared in the fold change.

Response: In this manuscript, the heatmap was employed to reflect the gene expression level of one: one pair of orthologous genes in V. fordii (Vf) and V. montana (Vm). The FPKM value of each gene was used to represent the gene expression level and the fold change was not compared here. The scale represented the FPKM value.

3. -line 448 "which suggesting"

Response: In this revised manuscript, we checked and corrected the grammar, such as "which suggesting" was changed to “which suggested”. Please refer to line 335 and line 365, page 10 and 11.